# Genome-Wide Analysis of the Growth-Regulating Factor Family in Peanut (*Arachis hypogaea* L.)

**DOI:** 10.3390/ijms20174120

**Published:** 2019-08-23

**Authors:** Kunkun Zhao, Ke Li, Longlong Ning, Jialin He, Xingli Ma, Zhongfeng Li, Xingguo Zhang, Dongmei Yin

**Affiliations:** College of Agronomy, Henan Agricultural University, Zhengzhou 450002, China

**Keywords:** peanut, GRF, gene family, expression analysis

## Abstract

Growth-regulating factors (GRFs) are plant-specific transcription factors that perform important functions in plant growth and development. Herein, we identified and characterised 24 *AhGRF* genes in peanut (*Arachis hypogaea*). *AhGRF* family genes were divided into six classes with OLQ and WRC domains. Transcriptome expression profile showed that more *AhGRF* genes, such as *AhGRF*5a gene, were at higher expression during pod development in *Arachis monticola* than cultivated species, especially at the pod rapid-expansion stage. *AhGRF*5a and *AhGRF*5b genes expressed at higher levels in pods than roots, leaves and stems tissues, existing in the difference between *Arachis monticola* and H8107. Exogenous GA3 application can activate *AhGRF*5a and *AhGRF*5b genes and H8107 line showed more positive response than *Arachis monticola* species. These results imply that these two *AhGRF* genes may be active during the peanut pod development.

## 1. Introduction

Transcription factors are highly variable and display functional diversity, and it is a DNA-binding protein that can specifically interact with cis-acting elements in the promoter region of eukaryotic genes. Through their interaction with each other and with other related proteins, it can activate or inhibit the transcription process, and it is the main regulator of gene expression. Plant growth-regulating factors (GRFs) play an important role in the regulation of plant growth and development [1]. GRFs were first discovered in rice (*Oryza sativa*); expression of *OsGRF*1 in *O. sativa* is increased significantly following gibberellin application, revealing its growth regulator function [2]. More recently, with advances in gene sequencing technology, the GRF family has been studied in many plant species including *Arabidopsis thaliana*, *O. sativa*, *Zea mays*, *Brassica napus*, *Cucumis sativus* L., *Nicotiana tabacum*, and other crops. Nine members have been identified in *A. thaliana*, compared with 12 in *O. sativa*, 14 in *Z. mays*, 17 in rape, 35 in cucumber, and 25 in tobacco [3,4,5,6,7]. GRF family proteins contain two conserved domains in the N-terminal region; QLQ (Glu-Leu-Glu, IPR014978) and WRC (Trp-Arg-Cys, IPR014977) [2,4,8]. In addition, most GRF proteins possess short-chain amino acids in the C-terminal region, for example, the TQL (Thr-Glu-Leu) and GGPL (Gly-Gly-Pro-Leu) motifs [9].

GRF proteins play crucial roles in various biological processes, molecular structure and expression levels in different tissues. GRFs are highly expressed in cell proliferation regions such as flowers, leaves and roots [10,11,12,13,14,15,16]. For example, *AtGRF*9 controls the development of leaves by negatively regulating the proliferation of leaf primordial cells [17]. Overexpression of *ZmGRF*1 increases the number of cells in leaves of *Z. mays*, as well as the size of leaves, while overexpression of *ZmGRF*10 reduces the number of palisade cells and decreases leaf size [18,19,20]. GRFs also perform regulatory roles in biological and abiotic stress [8]. GRF transcription factors are involved in seed development regulation, such as *AtGRF*1 is closely related to the weight and size of seeds. Overexpression of *OsGRF*4 can increase grain yield [21,22,23]. Meanwhile GRFs regulate fruit development in tomato [24]. Studies have found that most GRFs are target genes of microRNA396 (miR396), which is involved in the growth and development of various plants along with miR396 [10,12,14,15,23,24,25].

Peanut (*Arachis hypogaea*) is an allotetraploid (AABB 2n = 4x = 40); the AA subgenome is derived from the diploid wild species *Arachis duranensis*, and the BB subgenome is derived from the diploid wild species *Arachis ipaensis* [26]. Cultivated peanut is one of the most economically important oilseed crops and further enhancing the yield and quality is a main goal of peanut breeding programs. Members of the GRF protein family play important roles in plant growth, grain size and stress responses. GRF gene families have been identified in many plant species, but have not yet been reported in peanut. Meanwhile, the genome sequence of two diploid wild ancestors, *Arachis monticola* and cultivated peanuts, were reported [27,28,29,30], and more and more function genes could be explored and applicated to the peanut breeding [31,32]. Here, 24 *AhGRF* genes were identified and analysed in terms of phylogenetic relationship, gene structure and expression patterns in various tissues. The results provide a foundation for further function on *AhGRF* genes in peanut.

## 2. Results

### 2.1. Summary of the AhGRF Gene Family in Peanut

A total of 24 *AhGRF* genes were identified in peanut, named *AhGRF*1 to *AhGRF*20 based on their physical locations on chromosomes (Appendix A). The 24 *AhGRF* genes are distributed on 16 chromosomes. The chromosomes 01, 03, 08, 09, 10, 11, 13, 17, 19 and 20 have only one gene distribution. Two genes on chromosomes 2, 5, 6 and 15, and chromosomes 12 and 16 have three genes, respectively.

The *AhGRF* genes vary in length from 1365 bp (*AhGRF*1) to 6278 bp (*AhGRF*2a), with CDS lengths from 807 bp (*AhGRF*12b) to 1893 bp (*AhGRF*11). The number of exons also varies, from two in *AhGRF*1 to five in others. *AhGRF* genes encode proteins ranging from 268 (*AhGRF*12b) to 630 (*AhGRF*10) amino acids (aa), with an average length of 435 aa, and the molecular weights vary from 29.842 kDa (*AhGRF*12b) to 65.060 kDa (*AhGRF*6b). The isoelectric point (pI) ranges from 6.61 (*AhGRF*17) to 9.94 (*AhGRF*10), 21 *AhGRF* members pI > 7, while only *AhGRF*16c, *AhGRF*17 and *AhGRF*20 pI < 7. This may be related to their different roles in the peanut growth and development (Appendix A).

### 2.2. Genes Structure, Conserved Domains and Phylogenetic Analysis of AhGRF 

To further investigate the structural characteristics of *AhGRF* genes, we used NJ method (1000 bootstrap replicates) to construct an evolutionary tree for *AhGRF* protein sequences, which were divided into six classes (Figure 1).The genes of class II, III and IV have a similar structure and contain three or four exons, whereas class I (*AhGRF*1) has two exons and V (*AhGRF*12b) has five exons. Most homologous pairs of genes share high similarity in terms of the length and number of exons/introns, and these features are highly conserved.

Conserved domains in *AhGRF* family members were predicted (Figure 2). Most *AhGRF* genes contain four to six conserved domains, such as WRC, QLQ, FFD (Phe-Phe-Asp) and GGPL (Appendix A). *AhGRF*2b, *AhGRF*12a and *AhGRF*12c possess two WRC domains, *AhGRF*5a and *AhGRF*5b contain similar motifs (Appendix A). The specific distribution of the conserved motifs may lead to functional differences between the *AhGRF* genes.

The phylogenetic tree of GRF gene family members from four species was constructed based on full-length *AhGRF*, *AtGRF*, *OsGRF* and *GmGRF* protein sequences (Figure 3; Appendix A). A total of 68 GRFs were clustered into ten subgroups (I–X). *AhGRF* genes were distributed in seven groups, whereas subgroups III and VII have only *OsGRF* members. Subgroup VI has only *AtGRF* members. Among the subgroups, subgroup VIII was relatively small, with only one GRF. By contrast, subgroups I and IV contained the largest number of GRFs (six each), followed by subgroups V, IX and X have two GRFs and subgroup II (four). *AhGRF*5a and *AhGRF*15a were clustered together with *GmGRF*3-4, *GmGRF*4-3 and *AtGRF*5 in subgroup I. The phylogenetic tree suggested that the *AhGRF*s closer with *GmGRF*s and *AtGRF*s than *OsGRF*s, which may be because peanut, soybean and Arabidopsis are dicotyledonous plants.

### 2.3. Differential Expression Analysis of AhGRF Genes

To gain insight into the functions of *AhGRF* genes, we measured the expression levels of eighteen genes in three varieties during pod development and built a heatmap of the results (Figure 4A; Appendix A). More GRFs were expressed at higher levels in the wild species. *AhGRF*5a was expressed at high levels in *Arachis monticola* (A.mon) and H8107 species, and had highly expressed in the later stage of pod development. Meanwhile, *AhGRF*5a and *AhGRF*5b GFP expression vectors were constructed, and two proteins were localized on the nuclear (Figure 4B).

We further investigated the expression levels of *AhGRF*5a and *AhGRF*5b in different tissues between A.mon and H8107 (Figure 5), and found that genes expressed at higher levels in pod tissues than roots, leaves and stems tissues, existing in the difference between the two lines. Expression levels of *AhGRF*5a and *AhGRF*5b in leaves, roots and pods of A.mon were higher than H8107. In particular, expression of *AhGRF*5a and *AhGRF*5b in A.mon pods were 1.72-fold and 17.63-fold greater than in H8107, and *AhGRF*5b was achieved extremely significant differences. These results suggest that *AhGRF*5a and *AhGRF*5b may play the dominant role during the development of pods. 

### 2.4. Responses of AhGRFs to Exogenous GA3 Treatment

In order to investigate the response of *AhGRF*s to exogenous gibberellin A3 (GA3) application, we selected two-week-old seedlings spraying GA3 with 100 uM. Exogenous GA3 application can activate *AhGRF*5a and *AhGRF*5b genes and H8107 line showed more positive response than A.mon line (Figure 6). The expression levels of *AhGRF*5a and *AhGRF*5b were initially increased, then decreased and achieve highest level at 12 h in two lines. Thus, *AhGRF*5a and *AhGRF*5b may act as response factors to GA treatment.

## 3. Discussion

Herein, we identified 24 *AhGRF*s from peanut and explored their characteristics, further clustered into ten classes based on a phylogenetic tree of 68 GRF homologs from four representative plant species. Expression profiles showed that *AhGRF*5a was expressed at high levels in A.mon and H8107, and their expression levels revealed that *AhGRF*5a and *AhGRF*5b expressed higher levels in pod and showed positive response with GA3 treatment. 

Early expansion of the GRF gene family can be linked to whole-genome triplication that occurred in the common ancestor of eudicots, and further expansion in this family has occurred through several independent whole-genome duplications in various plant lineages [8]. Interestingly, early study identified 23 *AhGRF* genes in *Arachis duranensis* and *Arachis ipaensis* (Appendix A). Further analysis found *AhGRF*2-3 of wild peanut as *AhGRF*12a and *AhGRF*12b in cultivated species. This result may be due to gene expansion during evolution. Previous studies found that GRF genes may exhibit structural and functional differentiation in monocotyledonous and dicotyledonous plants [8], the phylogenetic tree suggested that the *AhGRF*s were closer with *GmGRF*s and *AtGRF*s than *OsGRF*s, which may be because peanut, soybean and Arabidopsis are dicotyledonous plants. Many GRFs are generally expressed at higher levels in actively growing tissues than in mature tissues, and it also play a role in regulating plant senescence [3,33,34]. GRFs regulate fruit development in tomato and overexpression of *BnGRF*2a results in increased seed weight and oil content [11,24]. High expression of *OsGRF*4 modulates tissue and organ size, resulting in larger grains, and enhanced grain yield [4,22,35,36]. Peanut pod development can be divided into two stages: pod expansion and pod filling. In general, the first sign of pod development is seen at 15~DAF (days after flowering), and pods enlarge to reach their maximum size at 35~DAF, at which point typical fruits are produced. Peanut pods mature at 60~DAF [37]. In this study, *AhGRF*5a and *AhGRF*15a were clustered together with *GmGRF*3-4, *GmGRF*4-3 and *AtGRF*5. The similarities are 63.27%, 59.02% and 28.45%, respectively. Meanwhile, *AhGRF*5a expression levels in pods were much higher than in other tissues, and expression analysis showed that *AhGRF*5a was expressed at high levels during at the pod rapid-expansion stage. This suggests that *AhGRF*5a may play important roles during seed formation. *Knotted*1*-like homeobox* (KNOX) is one of the most important regulators in the development and function of meristematic tissues, where it controls meristem development and restricts cell differentiation. GRFs are upstream repressors of KNOX genes that inhibit GA biosynthesis [38]. *BrGRF* genes expression is enhanced by GA3 treatment in Chinese cabbage [39]. Here, we measured the expression levels of *AhGRF*5a and *AhGRF*5b in two peanut varieties with GA3 treatment, and found that exogenous GA3 application can activate *AhGRF*5a and *AhGRF*5b genes. Thus, *AhGRF*5a and *AhGRF*5b act as response factors to GA3. And 12 h after GA3 application is an important time node. It can be inferred that GA3 treatment stimulates the expression of KNOX, and high KNOX levels subsequently lead to up-regulation of *AhGRF*5a and *AhGRF*5b, which was consistent with previous study in tobacco [40].

Our results provide a foundation for further function of *AhGRF* genes in peanut. Studies have found that most GRFs are target genes of miR396, which is involved in the growth and development of various plants, but the relationship of miR396 and *AhGRF*s needs to be explored. We found that *AhGRF*5a has a high expression in pod. In addition, plant circadian rhythm and other factors may also have an impact on gene response. This present study provides a foundation for a better understanding of the roles of *AhGRF*5a and *AhGRF*5b genes. In the future, the results need to be verified widely and potential function be explored in peanut.

## 4. Materials and Methods

### 4.1. Analysis of Peanut AhGRF Gene Family Members 

The sequences (genomic, CDS and amino acid) and physical locations were downloaded from peanutbase (http://www.peanutbase.org/). SMART (http://smart.embl-heidelberg.de/) was used to confirm the conserved QLQ and WRC domains. Molecular masses of putative *AhGRF* proteins were calculated using the compute pI/Mw tool in ExPASy (https://web.expasy.org/protparam/).

### 4.2. AhGRF Gene Structure and Phylogenetic Analysis

Protein sequence motifs were identified using the MEME program (http://meme-suite.org/). *AhGRF* gene structures were deduced using the Gene Structure Display Server (GSDS2.0, http://gsds.cbi.pku.edu.cn/index.php). Gene chromosomal location mapping was performed using MapGene2chrom webv2 (http://mg2c.iask.in/mg2c_v2.0/). Arabidopsis (http://www.arabidopsis.org/), *O. sativa* (http://rice.plantbiology.msu.edu/) and soybean (http://plants.ensembl.org/index.html) GRF protein sequences were downloaded from the appropriate databases, and sequence alignment was conducted with DNAMAN (Lynnon Biosoft Inc, San Ramon, CA, USA) software. An unrooted phylogenetic tree was constructed by MEGA5.2 software using the neighbour-joining (NJ) method with 1000 bootstrap replicates.

### 4.3. Expression Profiling Based on RNA Sequencing (RNA-Seq) Data

Three varieties, A.mon (tetraploid wild species) and recombinant inbred lines (RILs) H8106 and H8107, were selected. The main difference between the two RILs is the pod size: H8106 has medium-sized pods (3.2 cm long1.3 cm wide), and a 100-seed weight of 100 g, while H8107 has super-large pods (5.5 cm 2.07 cm) with a corresponding 100-seed weight of 182 g. The developmental stages of seeds 15~DAF, 25~DAF, 35~DAF, and 45~DAF were showed in Appendix A. Total RNA was extracted and used to purify poly (A) mRNA using Oligotex mRNA midi prep kit (QIAGEN, Germany). Sequencing libraries were generated using NEBNext UltraTM RNA Library Prep Kit for Illumina (New England Biolabs, Ipswich, MA, USA). Raw sequences were transformed into clean reads after data processing. These clean reads were then mapped to the reference genome sequence. Quantification of gene expression levels were estimated by fragments per kilobase of transcript per million fragments (FPKM). The expression of the *AhGRF*s family genes in these three different varieties at the different developmental stages was obtained. A heat map of *AhGRF* genes was generated using an R script based on normalised reads FPKM values of all genes transformed to log2 (value + 1).

### 4.4. Plant Materials and Hormone Treatments

A.mon and H8107 were used for transcriptome expression analysis. To verify the expression patterns of *AhGRF* genes, root, stem, seedling leaf (five leaf stage) and pod (45 DAF) samples were collected, immediately frozen in liquid nitrogen and stored at −80 °C until RNA extraction. For hormone treatments, seeds were cultivated with 1/2 Hoagland solution and grown with a 16/8 h light/dark photoperiod and 60% relative humidity at 32/25 °C, respectively. Two-week-old seedlings grown on plates were treated by spraying 100 μM GA3. Leaves of seedlings were selected before GA treatment and designated as GA-control check (GA-CK). Leaves of seedlings were collected at 1, 6, 12 and 24 h after GA3 treatment, immediately frozen in liquid nitrogen, and stored at −80 °C until RNA extraction.

### 4.5. Total RNA Extraction and cDNA Synthesis

RNA was extracted from roots, stems, leaves and pods using a DNAprep Pure Polysaccharide Polyphenol Plant Total RNA Extraction kit (TianGen, Beijing, China). RNA concentration and quality were evaluated with a NanoDrop One spectrophotometer (Thermo Fisher Scientific, Madison, WI, USA) and visualised by standard agarose gel electrophoresis (1%, *w*/*v*). Total RNA was then treated with DNAse to remove contaminating genomic DNA. First-strand cDNA was synthesised from 400 ng of DNA-free total RNA using PrimeScript RT Master Mix (Perfect Real Time) with oligo (dT) 20 primer following the manufacturer’s instructions (TaKaRa, Dalian, China).

### 4.6. Real-Time Quantitative PCR

Using gene-specific primers designed with the NCBI (https://www.ncbi.nlm.nih.gov/), qRT-PCR was carried out using TB Green Premix Ex Taq II (Tli RNaseH Plus) mix (TaKaRa) on a CFX96 Touch Real-Time PCR System (Bio-Rad, Hercules, CA, USA). Specific primers for GRF genes and actin (the housekeeping gene) were designed to amplify about 200~bp [32]. The reaction mixture included 10 μL TB Green mix, 1 μL forward primer, 1 μL reverse primer, 2 μL cDNA template and 6 μL nuclease-free H_2_O in a final volume of 20 μL. Thermal cycling included an initial denaturation at 95 °C for 30 s, followed by 40 cycles at 95 °C for 5 s and 60 °C for 30 s. The specificity of PCR amplification was monitored by melting curve analysis from 65 °C to 95 °C at 0.5 °C/s. After each PCR run, a dissociation curve was generated to confirm the specificity of the product. Three biological replicates with three technical replicates were performed for each reaction. The 2^−ΔΔCT^ method was employed to calculate the relative expression levels of *AhGRF* genes. Sequences of primers used for qRT-PCR are listed in Appendix A. 

### 4.7. Construction of AhGRF Transient Expression Vectors and Subcellular Localization Studies in Tobacco

To investigate the subcellular localization of the *AhGRF* proteins, they were transiently expressed as translational GFP (green fluorescent protein) fusion proteins in tobacco (*Nicotiana benthamiana*) leaf epidermal cells. The full-length coding sequences of *AhGRF*5a and *AhGRF*5b were amplified using Q5 high fidelity enzyme (New England Biolabs, Beijing, China) and cloned into ZT4 Blunt Fast Cloning Kit (Zoman Biotechnology, Beijing, China) according to the instruction of manufacturer (Appendix A). Then we designed two forward primers *AhGRF*5a-1 and *AhGRF*5b-1 containing a homologous recombination sequence (F:5′-ACAAATCTATCTCTCTCGAG-3′ R:5′-GCTCACCATGGATCC-3′) of the vector. The amplification products were using SE Seamless Cloning and Assembly Kit (Zoman Biotechnology, Beijing, China) ligated into the PFGC5941-35S-GFP (35S-GFP) vector. The recombined plasmids were then transformed into *Agrobacterium tumefaciens* strain EHA105. *Agrobacterium* transient expression and infiltration was carried out according to previously published protocols [41,42]. Leaves transformed with the 35S-GFP vector alone were used as controls. Two days after infiltration, fluorescence and bright-light images of transiently infected tobacco leaves were obtained using a Laser Scanning Confocal Microscopy (LSM710, Axio Obseror Z1, Zeiss, Germany). The primers used are listed in Appendix A.

## 5. Conclusions

In this study, 24 *AhGRF* genes were identified from the peanut genome, which were divided into six classes with OLQ and WRC domains. *AhGRF* genes were at higher expression during pod development in *Arachis monticola* than H8107. Exogenous GA3 can activate *AhGRF*5a and *AhGRF*5b genes, and cultispecies showed more positive response than tetraploid wild species.

## Figures and Tables

**Figure 1 ijms-20-04120-f001:**
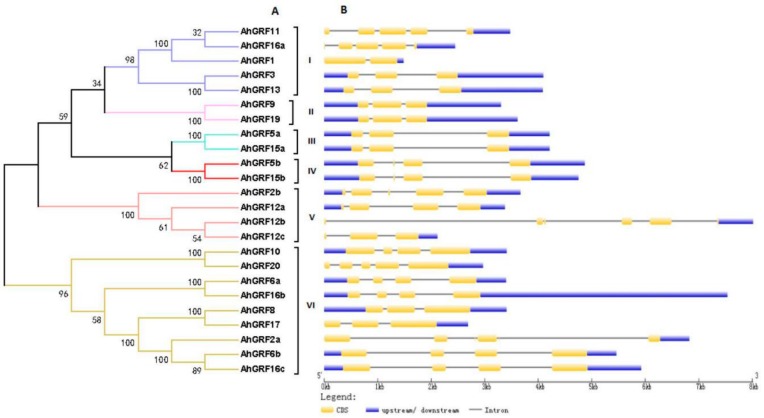
Exon–intron structures of *AhGRF* genes according to their phylogenetic relationships. (**A**) An unrooted phylogenetic tree was constructed based on the amino acid sequences. (**B**) Exon–intron structure analysis of *AhGRF* genes performed using the online tool GSDS. The lengths of exons and introns of each *AhGRF* gene are drawn to scale.

**Figure 2 ijms-20-04120-f002:**
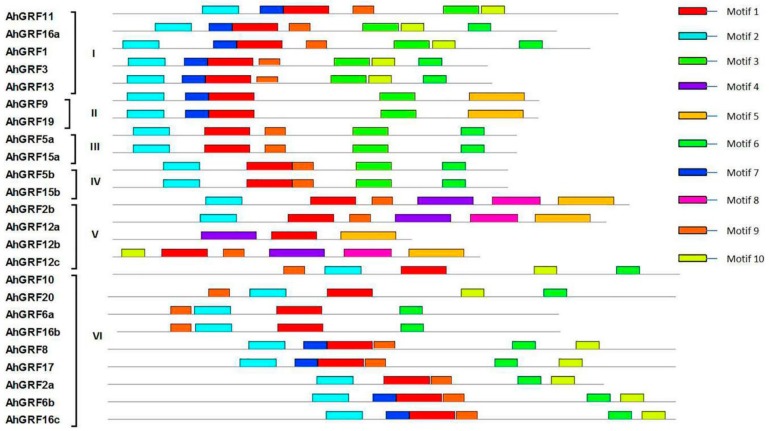
Distribution of conserved domains in *AhGRF* genes. Ten conserved motifs labeled with different colors were found in the *AhGRF* sequences using the MEME program. Among them, motif 1, motif 2, motif 4 and motif 6 are the QLQ, WRC, GGPL and FFD conserved domains. The sizes of motifs are proportional to their sequence length.

**Figure 3 ijms-20-04120-f003:**
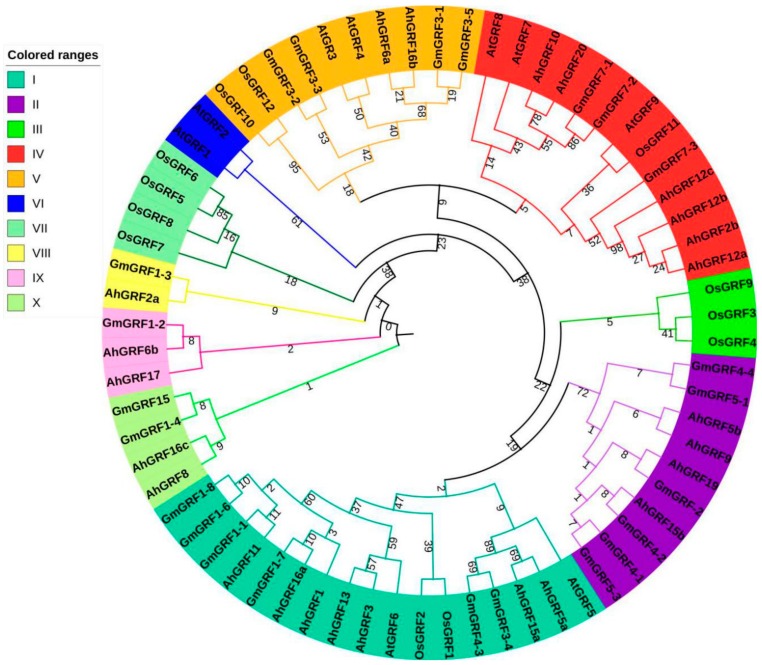
Phylogenetic tree based on the growth regulating factors (GRFs) protein family in four plants. *Ah*, peanut; *At*, *Arabidopsis thaliana*; *OS*, rice; *Gm*, soybean. The number stands for the confidence of the branch.

**Figure 4 ijms-20-04120-f004:**
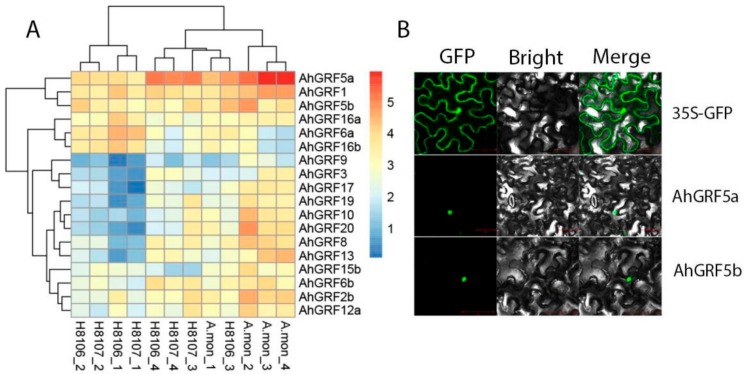
Heatmap and subcellular localization of *AhGRF* genes. (**A**): Heatmap of *AhGRF* genes expression in three peanut varieties. A.mon, H8106 and H8107 are peanut recombinant inbred lines (RILs). Pods from four developmental stages were used for expression analysis; 15~DAF (days after flowering) (−1), 25~DAF (−2), 35~DAF (−3) and 45~DAF (−4). Expression values from RNA-seq data were log_2_-transformed and are displayed as filled blocks coloured blue to red. (**B**): Subcellular localization of *AhGRF* proteins in tobacco leaves. *AhGRF5a* and *AhGRF5b* fusion proteins, as well as GFP alone. DAF, days after flowering.

**Figure 5 ijms-20-04120-f005:**
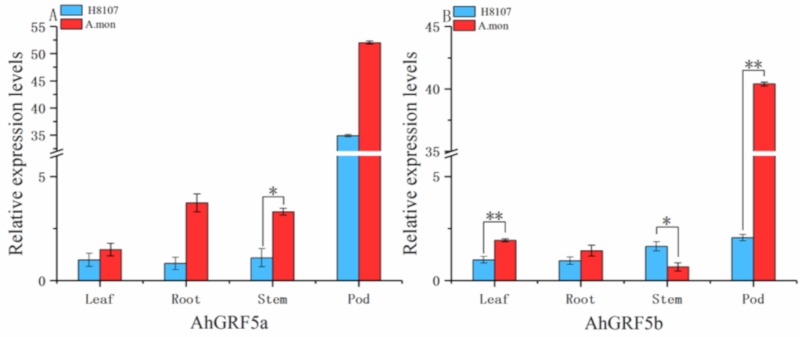
Expression analysis of *AhGRF* genes in different tissues. The qRT-PCR analysis of *AhGRF*s transcript levels was performed on leaves, roots, stems and pods. (**A**) *AhGRF*5a and (**B**) *AhGRF*5b. Experiments were repeated three times and vertical bars indicate standard errors. * and ** represent significant differences at *p* < 0.05 and *p* < 0.01, respectively.

**Figure 6 ijms-20-04120-f006:**
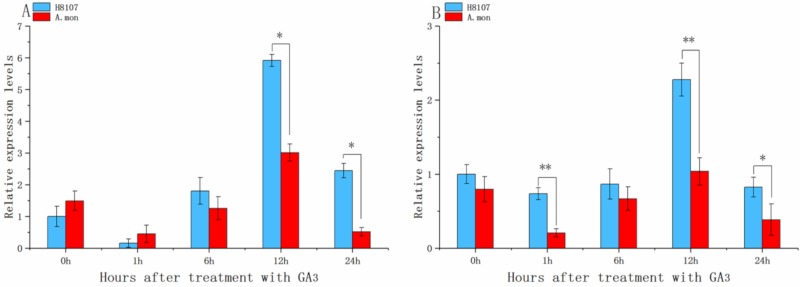
Expression analysis of *AhGRF* genes in response to treatment with the hormone GA3. qRT-PCR analysis of *AhGRF* transcript levels was performed on 2-week-old seedlings treated with GA3-0 h, GA3-1 h, GA3-6 h, GA3-12 h, GA3-24 h. (**A**) *AhGRF*5a and (**B**) *AhGRF*5b. Experiments were repeated three times and vertical bars indicate standard errors. * and ** represent significant differences at *p* < 0.05 and *p* < 0.01, respectively.

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
