# Peer review of "Genome-Wide Analysis of the Growth-Regulating Factor Family in Peanut (Arachis hypogaea L.)"

_ijms, 2019, doi:10.3390/ijms20174120_

Round 1
Reviewer 1 Report
The manuscript “Genome-wide analysis of the growth-regulating factor family in peanut” However, there still needs much work on polishing the language used in the manuscript. For example, L7 in abstract “acting” should be “active”; the first sentence in the introduction session should be restructured; L33-L34, what is the difference between “different in terms of tissue distribution” and “expression levels in different tissues”?; L55, “…for further function…”, function what?... Authors should spend more time on improving points like those above. The current manuscript is in a pretty low quality. Some figures and legends are just misleading. There are some additional suggestions for authors to improve the manuscript as below.
In result 2.1, the description of which chromosome has which number of AhGRF genes is very messy. Please try to restructure this sentence. In figure 1 legend, please mention what the number on each branch stands for? what the p-value stands for? Other than naming motif numerically, are there any actual name for those motifs? Figure 2A and Figure 1A are completely the same, why did you need a same information twice? I was surprised about the description in figure 3 legend, it is very hard to trace and please restructure them to make it clear. Also authors should label the meaning of number on each branch of this phylogenetic tree in figure 3. Authors should mentioned clearly how you processed the RNA-seq experiment and which reference genome you used. For RIL lines, which generations they are? Reads from different parental lines or remaining heterozygous sites may have strong influence on the alignment rate, and how did you handle this? The higher expression level in wild species may because of the alignment bias, authors should investigate that and discuss that. Figure 5, I could not find any different lower case in the figure; also authors should explain “relative expression levels” is relative to what? Figure 5 and 6 figure legend are totally messy, please spend enough time on editing those before submission. In figure 6, the significance level labeling is very suspicious, in subpanel B 24h, there is obvious difference, but the significance is only <0.05? please use the original data to recalculate that and provide the original data as a supplemental material. The expression pattern you interpreted in the result did not make any sense to me, based on this current figure, the only conclusion is when GA treated more than 12h, the difference is significant.
Reviewer 2 Report
The authors present analysis of gene family coding GRF transcription factors in Arachis.
Involvement of GRF genes in the regulation of variety of processes was extensively researched in many plant species. This paper presents first analysis of this gene family in Arachis with the emphasis on their role in pod development. The results have a preliminary character and can be used as a preface for further research.
Without a doubt the authors had put a lot of effort in both laboratory tests as well as bioinformatics analyses, however, they failed at presenting their results.
The title and abstract do not reflect the contents of the manuscript.
The authors studied the GRF gene family, claiming that they are important in the pod development in several varieties of Arachis. It is unclear however, on what basis the Arachis varieties were chosen. Do they have some significant difference in pod development?
Moreover, as a control for expression analysis of selected sequences after the application of GA3 not only ‘hour 0’ should be considered, but also parallel non-treated plants collected at the same time intervals. This should have been done in order to prevent false-positives caused by, for example, circadian rhythm, especially since the increased level of expression is observed after 12h, while in [40] it occurs after 1h. The authors should take into consideration the results from [40] and respond to them by adding adequate control.
In the materials and methods section, there is no information about the manner in which transcriptome analyses were carried out (library construction, sequencing method, number of repetitions, information about the origin of analyzed sequences etc.). This is a serious oversight.
The figures should be numbered in the order of the appearance in the manuscript, yet they are starting from Fig S2 and Table S2.
In the second paragraph of subsection 2.1 there is no reference to any figure/table.
The figures 1 and 2 should be inserted into manuscript in different order. It is more logical to first describe the gene (introns, exons etc.) and then the protein structure.
What is the reasoning behind selection of these particular plant species (O. sativa, G. max, A. thalinana) for the tree construction? Why not use more/different species? This should be clearly explained and substantiated. The results of these analyses also suggest, that further investigated GRF5a and b are specific only to Fabacae, is that true? Is the data on the function of homologues of GRF5 in G. max available? These points could also be discussed.
In the 2.3 section suddenly terms “A. mon” and “H8107” appear without any prior explanation – this needs to be corrected.
As I mentioned before there is no information about the basis of experiment used to create a heatmap (RNA-Seq?), there is also no explanation why only three varieties (and why these particular ones) were chosen. Moreover, in the figure 4 caption there are 5 varieties mentioned, yet in the text there are only three, as mentioned before.
The captions for figures 3 and 4 are clearly switched between them.
The attempt to describe the results from transcriptome analysis, qRT-PCR and cellular localization (additionally each one in different varieties) in a manner that is clear and understandable for the reader, in 12 lines of text, has not been successful. This section needs to be expanded and refined.
Figure 1. What is the meaning of question marks near the “p-value” and “motive localization”? (This should be removed) The localization of “A” and “B” should be corresponding to figure 2.
In the caption of figure 3 there are abbreviations for four plant species used in the tree creation, and then only the GRF genes descriptions from Glycine max – why only for this species? The descriptions of all sequences used for tree creation should be in the supplementary data.
In the caption of figure 5 there is a mention of "lower case letter represent significant differences", yet on the figure there are only stars.
The authors focused on pointing out the differences in expression of GRF5a between two varieties. From my perspective this tendency is rather similar and the focus should be shifted in the direction of versatility of these changes among different varieties. There is no mention of this during the discussion, which might be the cause of the differences suggested by authors.
In the discussion the function of GRF is mentioned again, whereas the emphasis should be shifted to functions of GRF5 homologues in other plant species, as well as discussing it.
Subsection 4.3 in materials and methods – there is no source for the plant cultivation protocols of the 3 varieties (is it identical as in article [39] which is also co-authored by authors of this manuscript, where varieties H8107 i H8106 were used?)
Subsection 4.4 in materials and methods – the first sentence is incorrect – “transcriptional”?
In the literature I have found 7 articles regarding GRF and miR396, while in the manuscript only 2 of them are mentioned (15 and 25), and they are not the most representative.
Moreover I have the impression that references in the text do not match the ones in the literature. For example in the text there is "Peanut pods mature at ~60 DAF [38]." While in the literature under number 38 there is "Choi, D., Kim, J.H., Kende, H. Whole genome analysis of the OsGRF gene family encoding plant-specific putative transcription activators in rice (oryza sativa l.). J. Plant and Cell Physiology. 2004, 45, 897-904. doi:10.1093/pcp/pch098." Which is an article about plant other than peanut, presumably this was meant to be reference to [39].
In the manuscript there is no section containing the list of supplementary files as well as statement about the possible conflict of interest.
Figure S1 shows the photograph of H8107 seedlings. What is the purpose? This would be sensible if all three varieties were photographed, along with the pod pictures from each variety.
Table S1: There is “action7” whereas it should be “actin7”
Table S4: In the description the word “genes” is doubled and there is no defined unit of expression stated in the transcriptome
Table S5 lacks the units (such as molecular weight). The purpose of giving two values in one column separated by “,” is unclear – this could have been done in two separated columns to make the datasheet more clear.
General remarks regarding the manuscript:
- the latin names should be written in italics
- the font size and style in the manuscript should be standardized (both in the text and in the figures), at least 4 different sizes were found, as well as different fonts used for figures.
- when multiple consecutive literature sources are cited, such as [10,11,12,13,14,15,16] it should be shortened to [10-16] to increase clarity.
Summary:
The manuscript requires a major, if not complete re-edition, including the addition of necessary information.
Round 2
Reviewer 1 Report
Please include details for RNA-seq data experiments in the methods part of your manuscript. Some English writing still needs to be improved, in the discussion session "We will focus on it ’ s function and regulation pathway in peanut.", there is no need to state what you are going to do in the future, this is a scientific paper but not a report. You can just describe what potentials people could analyze in the future based on your findings, also, it is "its" but not "it's".
Author Response
Please include details for RNA-seq data experiments in the methods part of your manuscript.
Response 1: Thanks for your suggestion. We have added some contents in materials and methods 4.3. The massive RNA-seq data is organized and will present in the future. This paper only showed the expression of the GRFs family genes.
4.3 Expression profiling based on RNA sequencing (RNA-Seq) data
Three varieties, A.mon (tetraploid wild species), and recombinant inbred lines (RILs) H8106 and H8107 were selected. The main difference between the two RILs is the pod size: H8106 has medium-sized pods (3.2 cm longÍ1.3 cm wide), and a 100-seed weight of 100 g, while H8107 has super-large pods (5.5 cmÍ 2.07 cm) with a corresponding 100-seed weight of 182 g. The developmental stages of seeds 15~DAF (days after flowering), 25~DAF, 35~DAF, and 45~DAF were showed in Fig. S3. Total RNA was extracted and used to purify poly (A) mRNA using Oligotex mRNA midi prep kit(QIAGEN,Germany). Sequencing libraries were generated using NEBNext UltraTM RNA Library Prep Kit for Illumina (NEB, USA). Raw sequences were transformed into clean reads after data processing. These clean reads were then mapped to the reference genome sequence. Quantification of gene expression levels were estimated by fragments per kilobase of transcript per million fragments (FPKM). The expression of the AhGRFs family genes in these three different varieties at the different developmental stages was obtained. A heat map of AhGRF genes was generated using an R script based on normalised reads FPKM values of all genes transformed to log2 (value + 1).
Some English writing still needs to be improved, in the discussion session "We will focus on it’s function and regulation pathway in peanut.", there is no need to state what you are going to do in the future, this is a scientific paper but not a report. You can just describe what potentials people could analyze in the future based on your findings, also, it is "its" but not "it's".
Response 2: Thanks for your suggestion. We have revised it. The modified results are as bellow.
In the future, the results need to be verified widely and potential function will be explored in peanut.
Reviewer 2 Report
Thank you for accepting my suggestions. I have no more comments.
Author Response
Thanks for your suggestion.